# Subject-Independent EEG Emotion Recognition Based on Genetically Optimized Projection Dictionary Pair Learning

**DOI:** 10.3390/brainsci13070977

**Published:** 2023-06-21

**Authors:** Jipu Su, Jie Zhu, Tiecheng Song, Hongli Chang

**Affiliations:** School of Information Science and Engineering, Southeast University, Nanjing 210096, China; jeep@seu.edu.cn (J.S.); jiezhu@seu.edu.cn (J.Z.); songtc@seu.edu.cn (T.S.)

**Keywords:** electroencephalogram (EEG), emotion recognition, projective dictionary pair learning, genetic algorithm, parameter optimization

## Abstract

One of the primary challenges in Electroencephalogram (EEG) emotion recognition lies in developing models that can effectively generalize to new unseen subjects, considering the significant variability in EEG signals across individuals. To address the issue of subject-specific features, a suitable approach is to employ projection dictionary learning, which enables the identification of emotion-relevant features across different subjects. To accomplish the objective of pattern representation and discrimination for subject-independent EEG emotion recognition, we utilized the fast and efficient projection dictionary pair learning (PDPL) technique. PDPL involves the joint use of a synthesis dictionary and an analysis dictionary to enhance the representation of features. Additionally, to optimize the parameters of PDPL, which depend on experience, we applied the genetic algorithm (GA) to obtain the optimal solution for the model. We validated the effectiveness of our algorithm using leave-one-subject-out cross validation on three EEG emotion databases: SEED, MPED, and GAMEEMO. Our approach outperformed traditional machine learning methods, achieving an average accuracy of 69.89% on the SEED database, 24.11% on the MPED database, 64.34% for the two-class GAMEEMO, and 49.01% for the four-class GAMEEMO. These results highlight the potential of subject-independent EEG emotion recognition algorithms in the development of intelligent systems capable of recognizing and responding to human emotions in real-world scenarios.

## 1. Introduction

EEG-based emotion recognition is a prominent research area in neuroscience and machine learning, as it holds the potential to enhance our understanding of the neural representation and objective measurement of emotions. Emotions play a crucial role in our daily lives, influencing decision making, behavior, and social interactions. However, traditional methods for measuring emotions, such as self-reporting or behavioral observation, possess limitations and are susceptible to bias. In contrast, an EEG offers a noninvasive, objective, and direct approach to measure emotions by detecting patterns of brain activity associated with specific emotional states [1]. This capability has the potential to advance the diagnosis and treatment of emotional disorders such as depression [2], anxiety [3], and post-traumatic stress disorder (PTSD) [4]. Moreover, the EEG holds promise for diverse applications, including human–computer interaction [5], affective computing [6,7], marketing research, and entertainment. Consequently, the development of reliable and accurate EEG emotion recognition systems bears great significance for the scientific community and society at large.

Numerous ongoing research efforts are dedicated to EEG emotion recognition, encompassing various research directions [8]. These include: (1) Time-Frequency Analysis, where the EEG signal is decomposed into different frequency bands using techniques such as wavelet transform or Fourier transform. Features extracted from each frequency band are employed to identify the emotional state [9,10]. (2) Independent Component Analysis (ICA) separates the EEG signal into independent components, with features extracted from each component used for emotion identification [11]. (3) Support Vector Machines (SVMs) are utilized as classifiers to categorize the EEG signal into distinct emotional states, employing features extracted from the signal [12]. (4) Fuzzy Logic models the uncertainty and imprecision in EEG signals for emotion recognition [13]. (5) Hidden Markov Models (HMMs) capture the temporal dynamics of EEG signals for emotion recognition [14]. (6) Multimodal emotion recognition combines EEG data with other modalities, such as facial expressions, speech, or physiological signals, to enhance the accuracy of EEG emotion recognition [15,16]. (7) Artificial Neural Networks (ANNs) are trained on the EEG signal to recognize different emotional states based on the extracted features [17]. (8) Deep learning techniques exhibit promising results in EEG-based emotion recognition, with ongoing exploration to develop models capable of capturing the intricate and dynamic patterns of brain activity associated with distinct emotional states [18,19,20,21]. (9) Transfer learning allows models trained on one dataset to adapt to another dataset with minimal retraining. Researchers are actively investigating the application of transfer learning to enhance the generalization performance of EEG emotion recognition models across different individuals and contexts [22,23].

Recent research has focused on developing algorithms capable of accurately classifying EEG signals associated with different emotional states. These algorithms employ machine learning techniques, such as support vector machines (SVMs) or deep learning, to assess the brain activity patterns within the EEG signal. However, a significant hurdle in EEG emotion recognition arises from the considerable variability in EEG signals across individuals and the challenges of controlling external factors that may influence brain activity, such as cognitive load or fatigue [22]. In order to address these challenges, we investigate methods to enhance the accuracy and resilience of classification algorithms.

The Projection Dictionary Pair Learning (PDPL) algorithm is a widely used machine learning technique employed in diverse applications, such as image processing [24], natural language processing [25], and Internet of Medical Things (IoMT) systems [26]. The PDPL serves as an effective approach for extracting significant features from high-dimensional data [27] and extends the standard Projection Dictionary Learning (PDL) algorithm. The fundamental concept of the PDPL revolves around identifying a pair of projection matrices that can map high-dimensional data to a lower-dimensional space while preserving the essential features. The algorithm comprises two main stages: dictionary learning and projection learning. In the dictionary learning stage, the algorithm acquires a set of basis functions or dictionary atoms that effectively and sparsely represent the input data by minimizing a cost function incorporating a data-fitting term and a sparsity-promoting term. In the projection learning stage, the algorithm determines a projection matrix that transforms the high-dimensional input data into a lower-dimensional space by minimizing a cost function containing a reconstruction error term and a regularization term. The PDPL process involves iterative updates of the dictionary and projection matrices until convergence. The PDPL offers several advantages over alternative machine learning algorithms, including improved sparsity, robustness to noise, enhanced performance, and interpretability, thereby rendering it suitable for a wide range of real-world applications.

The Projection Dictionary Pair Learning (PDPL) algorithm is a powerful and flexible method for learning dictionaries from high-dimensional data, offering advantages over traditional machine learning approaches. Its capacity to acquire interpretable and sparse dictionaries proves particularly valuable in applications that necessitate an understanding of the underlying features. The PDPL excels in selecting relevant features from datasets, enhancing the classification and clustering tasks. However, the effective tuning of its parameters relies on the researcher’s experience. In this study, we employ the Genetic Algorithm (GA), an optimization technique inspired by natural selection and evolution, to optimize the PDPL parameters for achieving optimal performance. The GA has demonstrated promising results in parameter optimization across various domains. For instance, the GA has been successfully employed to determine parameter values in the Deep Deterministic Policy Gradient, resulting in accelerated learning for the agent [28]. Additionally, the GA has been utilized for the multiobjective optimization of process parameters, employing a weighted objective sum method [29]. Moreover, the GA has been applied to SVM parameter optimization, effectively addressing grid search problems [30].

Genetic Algorithms (GA) simulate the natural selection process observed in biology, whereby individuals possessing favorable traits are more likely to survive, reproduce, and transmit their genes to subsequent generations [31]. In the context of optimization, these individuals represent potential solutions, while the genes correspond to the parameters or variables defining those solutions. GAs offer significant advantages in addressing intricate optimization problems, spanning domains such as engineering, finance, and artificial intelligence, where the search space is extensive, and alternative optimization algorithms prove ineffective [32].

The main contributions of this study can be summarized as follows:In recognition of the variability in EEG-based emotion recognition among individuals, we applied the PDPL algorithm to perform cross-subject analysis, with a specific focus on feature selection.The exploration of parameter space in the PDPL algorithm presents a substantial computational burden due to the wide range of parameter adjustments and the resulting extensive combinations. To address this challenge, we propose the utilization of the Genetic Algorithm (GA) for adaptive parameter optimization.Our proposed method surpasses conventional machine learning approaches, demonstrating exceptional recognition performance. Specifically, it achieves an average accuracy of 69.89% on the SEED database, 24.11% on the MPED database, 64.34% for the two-class GAMEEMO dataset, and 49.01% for the four-class GAMEEMO dataset. These results shed light on the effectiveness of emotion recognition, particularly for females, providing valuable insights into their emotional susceptibility.

In this academic paper, we address the issue of individual differences in EEG emotion recognition. To overcome this challenge, we introduce a novel PDPL algorithm based on genetic optimization. The proposed algorithm demonstrates superior recognition performance across subjects. The subsequent sections of the paper are organized as follows: Section 2 provides an overview of the materials and our proposed methodology, Section 3 presents the experimental results and corresponding discussions, and Section 4 concludes the paper while outlining future research directions.

## 2. Materials and Methods

### 2.1. EEG Emotion Database

To validate the effectiveness of our proposed method, we conducted experiments on three well-established multicategory EEG emotion databases. Specifically, we classified the datasets using both discrete models, such as a two-class model for GAMEEMO, a three-class model for the SJTU Emotion EEG Dataset (SEED Database), and a seven-class model for the Multimodal Physiological Emotion Database (MPED Database), as well as the dimensional model, employing a four-class model for GAMEEMO.

The SEED Database (https://bcmi.sjtu.edu.cn/home/seed/seed.html accessed on 18 June 2023) comprises emotional EEG signals collected from 15 subjects. For our experiment, we carefully selected a set of 15 Chinese film clips from a larger pool of materials based on their emotional valence (positive, neutral, and negative). These clips were meticulously edited to ensure coherence in eliciting emotions and to maximize emotional impact. Each clip had an approximate duration of 4 min. The experiment consisted of a total of 15 trials. Prior to the start of each clip, participants were provided with a 5 s hint. After viewing each clip, participants were given 45 s for self-assessment, followed by a 15 s rest period before proceeding to the next clip in the session. To assess their emotional reactions to the stimulus, participants were required to complete a questionnaire immediately after each clip viewing [33]. In our study, we utilized the dominant features provided in the database, specifically the differential entropy (DE) features, as the input for our proposed method.

The MPED Database (https://github.com/Tengfei000/MPED accessed on 18 June 2023) is an emotion database that comprises an EEG, galvanic skin response, respiration, and electrocardiogram signals. This comprehensive database encompasses data from 23 subjects, capturing 64 EEG channel signals and brain activity, while presenting 28 emotional stimulation videos. The videos encompassed a range of emotions, including joy, funny, anger, fear, disgust, sadness, and neutrality. All data were meticulously collected within a controlled laboratory environment [16].

The GAMEEMO Database (https://data.mendeley.com/datasets/b3pn4kwpmn/3 accessed on 18 June 2023), a collection of EEG signals acquired during computer games, was obtained from 28 individuals using the portable and wearable 14-channel Emotiv Epoc+ EEG device. Each participant engaged in four different computer games (boring, calm, horror, and funny) for 5 min each, resulting in a total EEG data duration of 20 min per subject. To evaluate the emotional experience, participants rated each game using the Self-Assessment Manikin (SAM) form, which measures arousal and valence [34]. Based on the stimulus material, EEG emotion can be categorized into two types: Positive–Negative models and four types of Arousal–Valence models, as presented in Table 1.

The extraction of log spectral power features allows for a quantitative assessment of power distribution across different frequency bands, enabling the study of brain activity. In both the MPED and GAMEEMO databases, we extracted the log spectral power features from the EEG signals [35]. To perform this extraction, the EEG signals were filtered using an order-8 zero-phase IIR Butterworth filter in five frequency bands: delta + theta (1–8 Hz), alpha (8–12 Hz), beta (12–35 Hz), gamma-1 (35–70 Hz), and gamma-2 (70–100 Hz). The root mean square of the filtered signals within each frequency band was computed using nonoverlapping 1 s windows. Finally, the logarithm of the root mean square was calculated for each window and electrode.

### 2.2. GA-PDPL for EEG Emotion Recognition

The PDPL is a machine learning algorithm utilized for unsupervised feature learning and data representation. Its primary objective is to acquire a dictionary of basis vectors for representing features in a lower-dimensional space. Nevertheless, parameter adjustment in the PDPL relies on experience. To achieve the optimal parameter combination, we incorporated a genetic algorithm (GA) to optimize these parameters. This approach allowed us to enhance the effectiveness of the PDPL in EEG emotion recognition. Figure 1 illustrates the framework of the GA-PDPL algorithm. Upon inputting the training dataset into the system, the GA parameters were generated. Through several iterations of parameter optimization, the best parameter combination was determined and employed to train the PDPL model. Subsequently, the trained model was tested on the test set, ultimately yielding the emotional category of each test sample as the output.

#### 2.2.1. Descriminative Dictionary Learning (DDL)

After the EEG signal was preprocessed, and the features were extracted, a sample could be expressed as $f\in R^{(b\times c)\times 1}$, where *b* and *c* are the number of frequency bands and the number of electrodes of the EEG signal, respectively. In addition, the label corresponding to the sample with a total of *K* emotional classes could be expressed as y∈{1,2,3,…,k,…,K}. We denote by $F=\{F_{1},\ldots ,F_{k},\ldots ,F_{K}\}$ and $Y=\{y_{1},\ldots ,y_{k},\ldots ,y_{K}\}$ a set of training samples and training labels from *K* classes, respectively, where $F_{k}=\left[f_{1},f_{2},\ldots ,f_{n}\right]\in R^{p\times n}$ is the training sample set of class *k*, p=b×c, $y_{k}=\left[y_{1},y_{2},y_{3},\ldots ,y_{n}\right]\in R^{1\times n}$ is the training label set of class *k*, and *n* is the number of samples of each class. DDL methods focus on acquiring a proficient data representation model from F to address classification tasks by leveraging the class label information of training data. This can be formulated within the framework presented below:(1)$$\min_{D,A}{∥F-\mathrm{DA}∥}_{F}^{2}+\lambda {∥A∥}_{p}+\Psi \left(D,A,Y\right).$$
In the training model (1), the scalar constant λ≥0, synthesis dictionary D, and coding coefficient matrix A of F over D are utilized. The data fidelity term ${∥F-DA∥}_{F}^{2}$ ensures the representation ability of D, while the $\ell_{p}$-norm regularizer ${∥A∥}_{p}$ is imposed on A. Additionally, a discrimination promotion function Ψ(D,A,Y) is used to ensure the discrimination power of D and A.

#### 2.2.2. PDPL Model

Deep Learning (DL) methods exhibit variations in their approach to learning a dictionary and classifier for all classes. Some methods employ a shared dictionary, while others utilize a structured dictionary to enhance discrimination. However, these methods commonly rely on $\ell_{0}$ or $\ell_{1}$-norm sparsity regularizers for the coding coefficients. Unfortunately, such reliance on sparsity regularization leads to inefficiencies during both the training and testing stages. To address this issue, we propose the PDPL model, which extends the conventional DL model presented in (1). The PDPL model introduces a pair of discriminative synthesis and analysis dictionaries. Unlike other DL methods, the PDPL model does not necessitate the use of costly $\ell_{0}$ or $\ell_{1}$-norm sparsity regularizers. Instead, the coding coefficients can be explicitly obtained through linear projection.

The discriminative model in Equation (Equation 1) aims to train a synthesis dictionary D that can sparsely represent the signal F [24,27]. Unfortunately, obtaining the code A for this dictionary requires an expensive $l_{\mathrm{rnomm}\;}$ sparse coding process. To improve the efficiency, we instead found an analysis dictionary $P\in R^{mK\times p}$ that satisfied A=PF, enabling the highly efficient representation of F without the need for sparse coding. To accomplish this, we learned an analysis dictionary using the synthesis dictionary D, resulting in the following formulated model,
(2)$$\left\{P^{*},D^{*}\right\}=\arg\min_{P,D}{∥F-DPF∥}_{F}^{2}+\Psi \left(D,P,F,Y\right).$$
In the DPL model, the analysis dictionary P was used for the analytical coding of F, while the synthesis dictionary D was utilized for the reconstruction of F, with discrimination function Ψ(D,P,F,Y) applied throughout. To improve the model’s efficiency, structured synthesis and analysis dictionaries $D=\left[D_{1},D_{2},\ldots ,D_{k}\right]$ and $P=\left[P_{1},P_{2},\ldots ,P_{K}\right]$ were learned. Each sub-dictionary pair for class *k* is produced by $D_{k}\in R^{p\times m}$ and $P_{k}\in R^{m\times p}$. To ensure that samples from class *i* (where i≠k) are projected towards a null space with the structured analysis dictionary P, $P_{k}$ was designed accordingly. This was achieved by leveraging sparse subspace clustering, which has demonstrated that under certain incoherence conditions, signals can be represented by their corresponding dictionary. The formulated equation for this process is shown below,
(3)$$P_{k}F_{i}\approx 0,\forall k\neq i.$$
The structured synthesis dictionary D can also be utilized to reconstruct the data matrix F. Specifically, the sub-dictionary $D_{k}$ can efficiently reconstruct the data matrix $F_{k}$ from the projective code matrix $P_{k}F_{k}$. Therefore, the dictionary pair was utilized to minimize the reconstruction error,
(4)$$\min_{P,D}\sum_{k=1}^{k}{∥F_{k}-D_{k}P_{k}F_{k}∥}_{F}^{2}.$$
Based on the preceding discussion, the formulation of the DPL model can be expressed as follows,
(5)$$\begin{array}{c} \left\{P^{*},D^{*}\right\}=\arg\min_{P,D}\sum_{k=1}^{k}\left|\right|F_{k}-D_{k}P_{k}F_{k}{∥}_{F}^{2}+\lambda ||P_{k}{\bar{F}}_{k}{\left|\right|}_{F}^{2},\;s.t.\;{∥d_{i}∥}_{2}^{2}\leq 1.\; \end{array}$$

The synthesis dictionary is represented by matrix D, which consists of atoms denoted as $d_{i}$. To ensure stability in the Projection Dictionary Learning (PDL) process, the energy of each atom is constrained to prevent the trivial solution $P_{k}=0$. Additionally, the complement of $F_{k}$ in the entire training set F is denoted as ${\bar{F}}_{k}$. While sparse coding is not necessarily crucial for classification, the DPL model offers faster computation and demonstrates highly competitive classification performance. Therefore, the following approach was adopted for classification purposes. To optimize the nonconvex objective function in Equation (Equation 5), a variable matrix A was introduced, and Equation (Equation 5) was relaxed to the following problem:(6)$$\left\{P^{*},A^{*},D^{*}\right\}=\arg\min_{P,A,D}\sum_{k=1}^{K}{∥F_{k}-D_{k}A_{k}∥}_{F}^{2}=\left|\right|P_{k}F_{k}-A_{k}{∥}_{F}^{2}+\lambda ||P_{k}F_{k}{\left|\right|}_{F}^{2},\;s.t.\;{∥d_{i}∥}_{2}^{2}\leq 1.$$

The objective function in Equation (Equation 6) consists of terms involving the Frobenius norm, which is facilitated by a scalar constant τ, ensuring ease of solving. To initialize the analysis dictionary P and synthesis dictionary D, random matrices with a unit Frobenius norm were initially employed. Subsequently, the minimization process proceeded by iteratively updating A and D,P. The minimization procedure involved alternating between the following two steps:(1)Fix D and P, update A,
(7)$$A^{*}=\arg\min_{A}\sum_{k=1}^{K}{\left|\left|F_{k}-D_{k}A_{k}\right|\right|}_{F}^{2}+\tau {\left|\left|P_{k}F_{k}-A_{k}\right|\right|}_{F}^{2}.$$
We can obtain a closed-form solution for this standard least-squares problem:(8)$$A_{k}^{*}={\left(D_{k}^{T}D_{k}+\tau I\right)}^{-1}\left(\tau P_{k}F_{k}+D_{k}^{T}F_{k}\right).$$

(2)Fix A, update D and P,(9)$$\left\{\begin{array}{c} P^{*}=\arg\min_{P}\sum_{k=1}^{K}\left|\right|P_{k}F_{k}-A_{k}{\left|\right|}_{F}^{2}+\lambda ||P_{k}{\bar{F}}_{k}{\left|\right|}_{F}^{2}\\ D^{*}=\arg\min_{D}\sum_{k=1}^{K}\left|\right|F_{k}-D_{k}A_{k}{\left|\right|}_{F}^{2},\;s.t.\;{∥d_{i}∥}_{2}^{2}\leq 1. \end{array}\right.$$
We can obtain closed-form solutions for *P* as follows:(10)$$P_{k}^{*}=\tau A_{k}F_{k}^{T}\left(\tau F_{k}F_{i}^{T}+\lambda {\bar{F}}_{k}{\bar{F}}_{k}^{T}+\gamma I\right),$$
where γ is a small number, and I is the identity matrix. Introducing a variable S can optimize the D problem:(11)$$\begin{array}{c} \min_{D,S}\sum_{k=1}^{K}∥F_{k}-D_{k}A_{k}{\left|\right|}_{F}^{2},\;s.t.\;D=S,\;{∥S_{i}∥}_{2}^{2}\leq 1. \end{array}$$
The optimal solution of (11) can be obtained by the Alternating Direction Method of Multipliers algorithm:(12)$$\left\{\begin{array}{c} D^{\left(r+1\right)}=\arg\min_{D}\sum_{k=1}^{K}{∥F_{k}-D_{k}A_{k}∥}_{F}^{2}+\rho ||D_{k}-S_{k}^{\left(r\right)}+T_{k}^{\left(r\right)}{\left|\right|}_{F}^{2},\\ S^{\left(r+1\right)}=\arg\min_{S}\sum_{k=1}^{K}\rho ||D_{k}^{\left(r+1\right)}-S_{k}^{\left(r\right)}+T_{k}^{\left(r\right)}{\left|\right|}_{F}^{2},\;s.t.\;D=\left|\right|S_{i}{\left|\right|}_{2}^{2}\leq 1,\\ T^{\left(r+1\right)}=T^{\left(r\right)}+D_{k}^{\left(r+1\right)}-S_{k}^{\left(r+1\right)},\;\mathrm{update}\;\rho \;\mathrm{if}\;\mathrm{appropiate}. \end{array}\right.$$

The proposed Dictionary Pair Learning (DPL) model exhibits a fast training process owing to the rapid convergence of closed-form solutions for variables A and P in each optimization step. The optimization of D is based on the Alternating Direction Method of Multipliers (ADMM), which also demonstrates quick convergence by stopping iterations when the energy difference between two consecutive iterations is below 0.01. Upon convergence, the analysis dictionary P and synthesis dictionary D are obtained as outputs for classification. The objective functions presented in Equation (Equation 9) are designed to enhance the discriminative power of P while minimizing the reconstruction error. This balanced approach allowed the model to achieve both effective discrimination and strong representation capabilities.

During the classification stage, the residual values of samples within a class were utilized. The analysis sub-dictionary $P_{k}\left({}^{*}\right)$ produces small coefficients for samples that do not belong to class *k*, while the synthesis sub-dictionary $D_{k}\left(*\right)$ reconstructs the samples from class *k*. Consequently, the residual $\left|\right|F_{k}-D_{k}^{*}P_{k}^{*}F_{k}{\left|\right|}_{F}^{2}$ for a sample in class *k* is smaller than the residual $\left|\right|F_{i}-D_{k}^{*}P_{k}^{*}F_{i}{\left|\right|}_{F}^{2}$ for a sample not in class *k*. During the testing phase, the residual of an unknown query sample $f_{t}$ is computed for each class. The class associated with the minimum residual is assigned as the class for the testing sample. This testing process can be formulated as follows:(13)$$\mathrm{identity}\left(f_{t}\right)=\arg\min_{i}{∥f_{t}-D_{i}P_{i}f_{t}∥}_{2}.$$
If the minimum residual in Equation (Equation 12) corresponds to class *i*, then sample $f_{t}$ is assigned to class *i*, where $D_{i}$ and $P_{i}$ represent the synthesis and analysis sub-dictionaries for that class, respectively.

#### 2.2.3. The GA for the Parameter Optimization of the PDPL

The genetic algorithm (GA) is a widely used optimization technique inspired by the natural selection process in biology. It is commonly employed to find the optimal parameters for machine learning algorithms. Initially, the algorithm generates an initial population of potential parameter values, and each solution in the population is evaluated using a fitness function that assesses its performance on a given task. Solutions with higher fitness scores are given preference for selection in the next generation. Genetic operators, such as mutation and crossover, are then applied to the selected solutions to generate new offspring solutions. Mutation randomly alters some of the parameters in a solution, while crossover combines the parameters of two solutions to create a new one. The newly created offspring solutions undergo evaluation using the fitness function, and those with higher fitness scores are selected for the next generation. This process continues until a termination criterion is met, such as reaching a maximum number of iterations or convergence of the fitness function. Finally, the optimal parameter values for the given task are represented by the solution with the highest fitness score obtained at the end of the algorithm.

The parameter optimization algorithm flow of the GA-PDPL utilizes the Sheffield University genetic algorithm toolbox (gatbx).

Initialization: An initial population of solutions is generated by randomly assigning values to the parameters. The initialization parameters, which include the maximum genetic algebra, population size, crossover function, mutation probability, and t parameter of PDPL, are presented in Table 2. Furthermore, the GA optimization process involves tuning four PDPL parameters: *m*, τ, λ, and γ. The threshold ranges and coding methods for these parameters are provided in Table 3.Evaluation: The fitness of each solution in the population is assessed using the projection dictionary pair learning (PDPL) algorithm and a fitness function. In this study, we defined the fitness function as the accuracy of the PDPL recognition on the test set. The calculation method for the fitness is illustrated in Figure 2 and can be found in Formula (14). To incorporate the research background, which was unrelated to the subjects, we introduced a leave-one-out subject cross test into the fitness calculation. The final fitness value was determined by averaging the accuracy across all subjects.
(14)$$Fitness=\frac{1}{N}\sum_{i=1}^{N}{\mathrm{Accuracy}}_{i}.$$Selection: Choose a subset of solutions to serve as parents for the next generation based on their fitness scores.Crossover: Generate new solutions by combining the parameters of the selected parents through crossover.Mutation: Introduce random changes to the parameters of some solutions to explore different areas of the search space.Evaluation: Assess the fitness of the newly created solutions resulting from crossover and mutation.Replacement: Select the top-performing solutions from both the previous and new generations to form the subsequent generation.Termination: Stop the algorithm when a specified termination criterion is met, such as reaching the maximum number of generations or achieving the desired level of fitness.Output: Provide the best solution obtained by the genetic algorithm (GA), which corresponds to the optimal parameter values for the projection dictionary pair learning algorithm.

## 3. Results and Discussion

The experimental test protocol followed the widely adopted leave-one-subject-out cross-validation (LOSOCV) procedure, which ensures robustness across subjects. We thoroughly analyze and discuss the obtained experimental results, focusing on aspects such as recognition accuracy, parameter variations, sex disparities, and a comparison with the state-of-the-art (SOTA) method.

### 3.1. Recognition Results of the GA-PDPL Method on Three Databases

Accuracy, an essential metric for evaluating the effectiveness of the proposed method, represents the ratio of the correctly classified samples to the total number of samples in a given test dataset. To demonstrate the efficacy of the proposed optimization method, we conducted a comparison of the effects of PDPL on three databases without parameter optimization. In this comparison, we utilized the default values for the four PDPL parameters, namely DictSize=32, τ=0.03, λ=0.003, and γ=0.0001. The recognition results are presented in Table 4. Across the four experimental conditions, the GA-PDPL consistently outperformed the PDPL in terms of accuracy. Particularly noteworthy was the significant improvement achieved by the GA parameter optimization on the SEED dataset, where the accuracy was 18.87% higher compared to the non-optimized version.

The 95% confidence interval is a valuable tool for assessing the reliability and uncertainty of recognition results in machine learning models. It provides a range within which we can be 95% confident that the actual result falls when conducting multiple experiments or sampling. In this study, we calculated the confidence intervals for all experimental accuracies, both with and without the GA for parameter adjustment. The results are presented in Table 5. Statistical analysis reveals that the upper and lower limits of the confidence interval for the proposed GA-PDPL method exceeded those of the PDPL method, thus demonstrating the effectiveness of our approach.

To enhance the visualization of the recognition results for each subject in the test set, we created bar graphs to display the outcomes for the three databases. Figure 3 represents the SEED database, Figure 4 corresponds to the MPED database, Figure 5 pertains to the GAMEEMO (two-class) dataset, and Figure 6 corresponds to the GAMEEMO (four-class) dataset. The GA-PDPL method demonstrates a substantial improvement in the recognition performance for most subjects across the four experiments.

The bar graph in Figure 3 illustrates that, with the exception of subjects 5 and 10, the accuracy rates for all other subjects surpassed 60%, with subject 11 achieving close to 100%. The optimization of the GA parameters resulted in a significant improvement in accuracy for all subjects except subject 14. These findings indicate that the GA enhanced the performance of the PDPL, enabling a better characterization of the EEG’s emotional characteristics.

Among the 23 subjects in the MPED database, as depicted in Figure 4, subject 14 achieved the highest accuracy, followed by subjects 18 and 23. The remaining subjects exhibited accuracy rates ranging from 20% to 30%. These considerable variations in individual performance can be attributed to several factors. Firstly, EEG signals are susceptible to various sources of noise interference, such as electromagnetic and power frequency disturbances. Additionally, there exist substantial individual differences among subjects, including disparities in emotional cognition and EEG responses. However, apart from subjects 8 and 22, the optimization of the GA parameters significantly enhanced the accuracy for all other subjects. This outcome demonstrates the efficacy of the GA in improving EEG emotion recognition through parameter optimization in a multi-classification setting.

The proposed method yielded a notable improvement in the accuracy rate of the latter subjects in both the two-class and four-class experiments of the GAMEEMO dataset. A comparison between the two-category and four-category experiments reveals that the two-category experiment achieved better recognition performance, while the four-category experiment demonstrated more substantial overall improvement. These findings are illustrated in Figure 5 and Figure 6.

### 3.2. Parameter Optimization Analysis of the GA-PDPL Method

The optimization procedure enhanced the performance of the Projection Dictionary Pair Learning (PDPL) algorithm through the utilization of Genetic Algorithms (GAs). Its objective was to iteratively refine the model’s parameters and feature selection process, thereby maximizing the classification performance. By employing the GA optimization algorithms and statistical techniques, the procedure aimed to find the optimal configuration that minimized the errors or maximized the performance metrics, resulting in improved classification outcomes. The optimization procedure offers several benefits. Firstly, it enables the fine-tuning of model parameters, optimizing their values to better align with the underlying data. This refinement process enhances the model’s ability to capture intricate patterns and relationships within the data, thereby improving the classification performance. Secondly, the optimization procedure facilitates feature selection or feature weighting, allowing for the identification of the most informative features for the classification task. By prioritizing the relevant features and reducing the influence of the irrelevant or redundant ones, the procedure enhances the discriminative power of the classifier. Additionally, the optimization procedure helps address the challenge of overfitting, which is common in classification tasks. By optimizing the regularization parameters or employing techniques such as cross validation, the procedure prevents the model from excessively memorizing the training data. Instead, it encourages the model to generalize well to unseen data, leading to an improved generalization performance and a reduced likelihood of erroneous classifications on new instances.

Figure 7, Figure 8, Figure 9 and Figure 10 illustrate the optimization results of the four parameters of the Genetic Algorithm (GA) utilized in the Projection Dictionary Pair Learning (PDPL) algorithm, as well as the corresponding change curves of the GA fitness function with increasing iterations on the three databases. In Figure 7, the variations in the number of iterations led to fluctuations in the *m* parameter, a decrease in the final value of τ, an increase in γ, a significant decrease in λ, and a gradual increase in the fitness function. Figure 8 exhibits a similar pattern to Figure 7 for each parameter and the fitness function, except for a substantial reduction in the τ parameter. The change curve of the MPED database demonstrated more pronounced changes compared to the SEED, potentially due to the larger amount of data available for MPED subjects, facilitating improved model learning.

The parameter changes in the SEED and MPED databases reveal that as the number of iterations increased, smaller values of τ and λ within their respective ranges and larger values of γ led to higher fitness levels. The *m* parameter underwent a process of feature selection, exhibiting a minimal significant increase or decrease. This adaptive process is beneficial for accommodating new test sets or subjects. Additionally, the growth trend of the fitness function can be observed through its change curve. With sufficient computing power, increasing the number of iterations allowed the fitness function to continue growing, resulting in a further improvement in the accuracy rates on each database.

Interestingly, for the same GAMEEMO dataset, different classification objectives yielded distinct trends and rules for the four parameters. Thus, increasing or decreasing specific parameters may have varying effects on different datasets. The change modes and combinations of the four parameters also differed across each dataset. Consequently, Genetic Algorithms (GA) provide an effective method for simultaneously optimizing these four parameters.

### 3.3. Emotion Recognition Performance of the GA-PDPL Method with Regard to Sex

It is widely acknowledged that emotional processing differs between men and women. Previous studies have provided evidence supporting this claim, indicating that women tend to display more authentic emotional expressions, while men exhibit greater control over their enthusiasm [36]. Furthermore, research has revealed that distinct brain networks are engaged by men and women when processing sad, depressed, and humorous audiovisual stimuli [37]. Women typically express their emotions through appearance and interpersonal interactions, whereas men tend to express their emotions through activities [38]. Although these studies contribute to our understanding of sex differences in emotional processing, they lack objective evidence and quantitative assessment [39]. Therefore, this study aimed to investigate sex differences in emotional processing using a more rigorous approach.

To address this objective, data were collected from two databases, and the recognition results of subjects of different sexes were analyzed. The mean and standard deviation values were calculated for accuracy, as depicted in Figure 11. The study’s findings revealed that the women exhibited higher average accuracy rates than the men in both databases, suggesting superior emotional cognition abilities. Additionally, the men displayed smaller standard deviations compared to the women, indicating greater emotional stability. These findings contribute objective evidence and provide a quantitative assessment of sex disparities in emotional processing. They support the notion that men and women process emotions differently, with women demonstrating higher emotional cognition abilities and men exhibiting greater emotional stability. These distinctions could be attributed to variations in brain networks and socialization processes.

The present study offers empirical support for the existence of sex differences in emotional processing. These findings emphasize the importance of considering sex when investigating emotional processing and have implications for enhancing communication and interpersonal relationships. Further research is warranted to explore the underlying mechanisms driving these sex disparities and their potential impact on mental health and wellbeing.

### 3.4. Training and Testing Time of the GA-PDPL

The genetic algorithm can be time-consuming when optimizing parameters, whereas the PDPL method offers the advantage of speed. In the context of the emotional brain–computer interface, the recognition time for the samples is a crucial factor. Thus, we recorded the training and testing time of the model on the test device (MATLAB 2019b, Intel(R) Core(TM) i5-9600KF CPU with 32.0 GB RAM). Considering the variations in sample size, stimuli, and categories across the three databases, we calculated the testing time for each sample, as presented in Table 6. Observing the results, although the time required for parameter optimization and model training was considerable, the testing time for each sample amounted to only a few thousandths of a second. This rapid calculation speed holds promising prospects for real-time detection of emotional changes in future real scenarios. In comparison to the PDPL method, the training time was longer, but the testing time was significantly reduced. Notably, the testing time for the GAMEEMO dataset was the shortest. This difference may stem from the dataset’s fewer signal channels, leading to a reduced number of features and consequently faster model recognition.

### 3.5. Comparison of the GA-PDPL Method and SOTA Method

We compared our proposed GA-PDPL method with state-of-the-art (SOTA) approaches in subject-independent EEG emotion recognition settings using the SEED, MPED, and GAMEEMO datasets. The results are presented in Table 7, Table 8, and Table 9, respectively. The compared algorithms were classic machine learning algorithms, which we applied to our experimental data and experimental results obtained under the same test protocol. Since the methods on the GAMEEMO dataset have not undergone subject validation, we included the classic method as a comparative approach in this experiment. From the tables, it can be concluded that our proposed method outperformed the current conventional methods in subject-independent protocols. Compared to the KLIEP [40], ULSIF [41], STM [42], SVM [43], KPCA [44], TCA [45], KNN [46], Random Forest [47], PDPL [27], and SA [43], our method utilized a comprehensive dictionary and an analysis dictionary to enhance the feature representation. Additionally, we employed a genetic algorithm (GA) for parameter optimization to select the optimal dictionary and parameters, thus achieving the best recognition performance. Furthermore, on the MPED database, our method outperformed two deep learning methods, DANN [48] and A-LSTM [16]. Deep learning methods exhibit great power, especially in parameter learning, but for small-sample datasets such as EEG emotions, deep learning models are prone to overfitting. To mitigate overfitting to some extent, our method reduced the dimensional space of the features to enhance the model training.

## 4. Conclusions

This paper presented a subject-independent EEG emotion recognition method employing genetically optimized projection dictionary pair learning. The experimental results demonstrated that the proposed method surpasses existing traditional machine learning algorithms. Moreover, our findings indicated superior recognition performance on female subjects. These outcomes hold implications for practical applications in emotional brain–computer interface devices. Nevertheless, this research had certain limitations. Due to computational constraints, we were unable to explore larger parameter settings within a limited timeframe, particularly in terms of the number of iterations, which resulted in a restricted optimization range. Future research should prioritize subject-independent and cross-database EEG emotional recognition. Such investigations have the potential to advance the development of emotional brain–computer interface systems applicable in real-world scenarios. 

## Figures and Tables

**Figure 1 brainsci-13-00977-f001:**
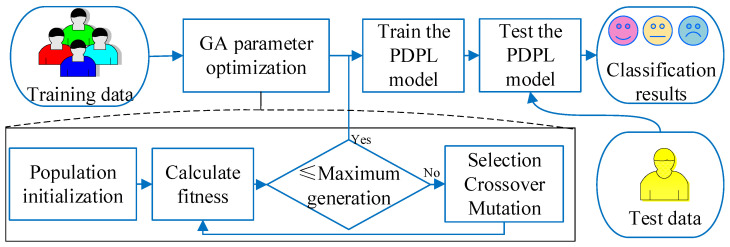
Framework of the GA-PDPL algorithm for EEG emotion recognition.

**Figure 2 brainsci-13-00977-f002:**
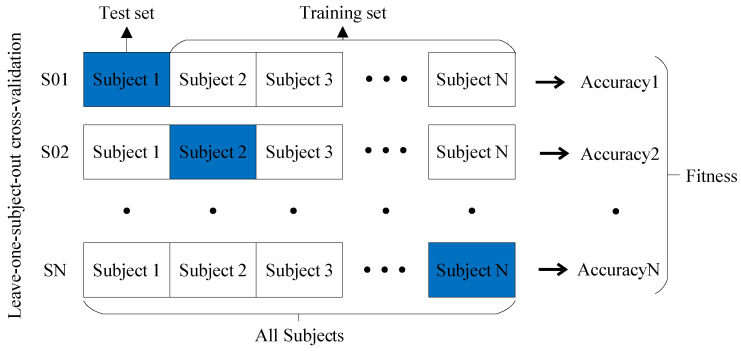
Leave-one-subject-out cross validation and fitness.

**Figure 3 brainsci-13-00977-f003:**
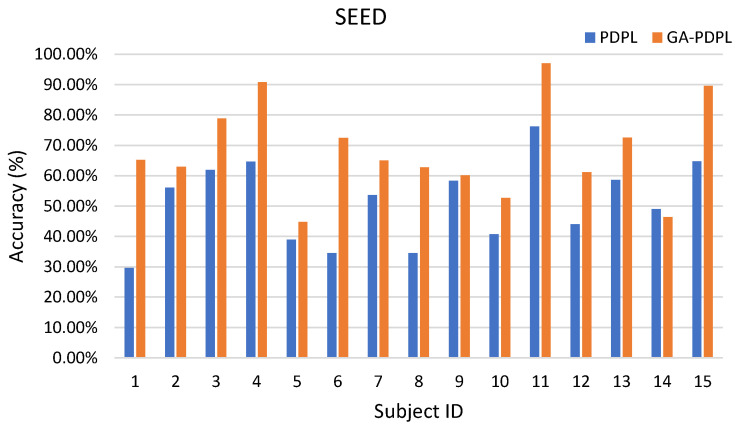
Accuracy (%) of each subject in the SEED database.

**Figure 4 brainsci-13-00977-f004:**
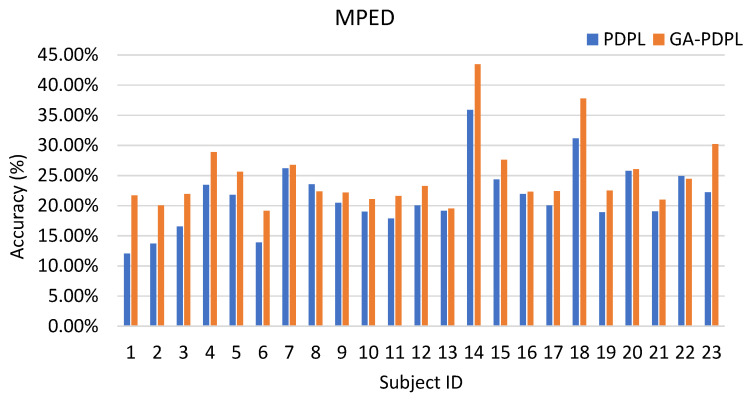
Accuracy (%) of each subject in the MPED dataset.

**Figure 5 brainsci-13-00977-f005:**
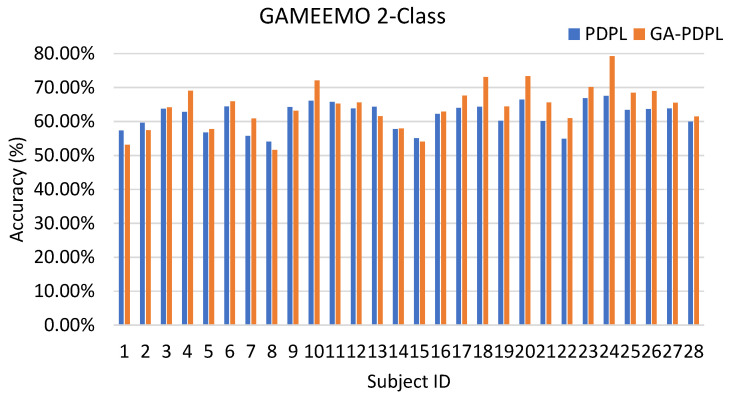
Accuracy (%) of each subject in the GAMEEMO (two-class) dataset.

**Figure 6 brainsci-13-00977-f006:**
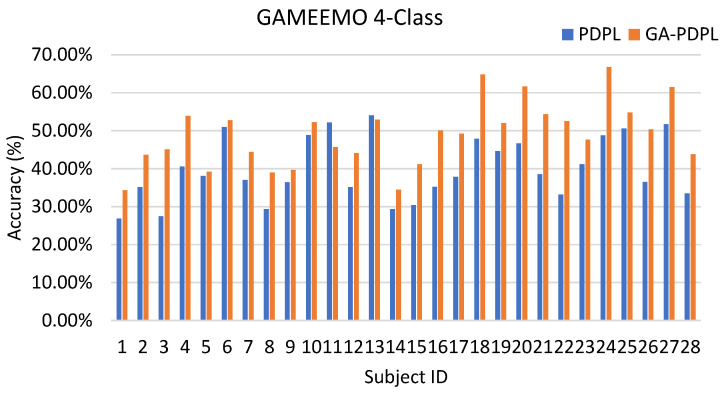
Accuracy (%) of each subject in the GAMEEMO (four-class) dataset.

**Figure 7 brainsci-13-00977-f007:**
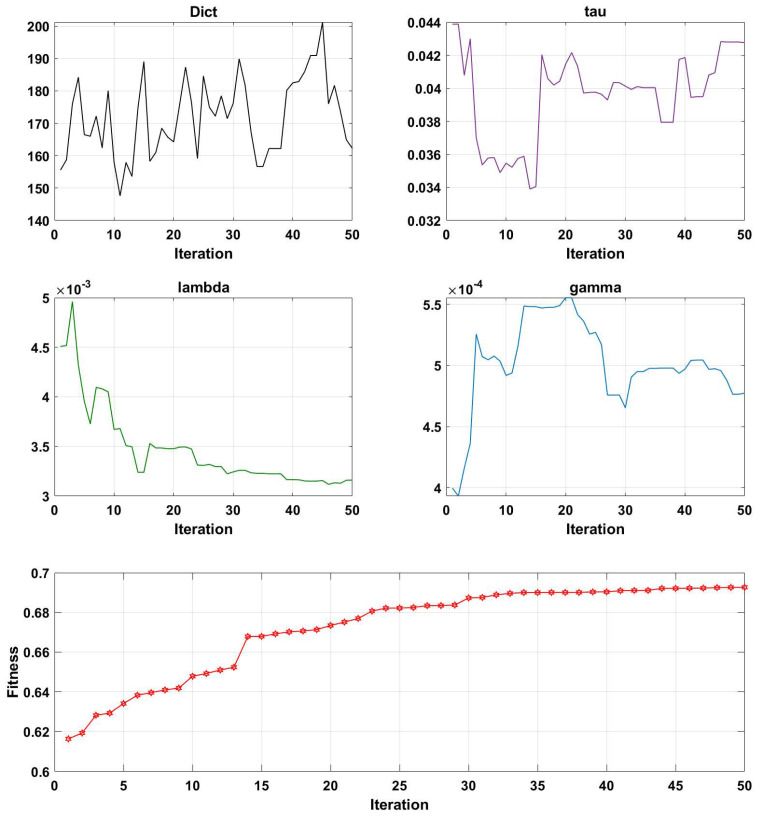
Change curves of the four parameters (*m*, τ, λ, and γ) and the fitness function on the SEED dataset. Here, ’Dict’ represents the *m* parameter. The first four line charts are the optimized values of the PDPL parameter optimization curve based on the GA at different iterations. The last line graph is the change curve of the fitness function as the number of iterations increases.

**Figure 8 brainsci-13-00977-f008:**
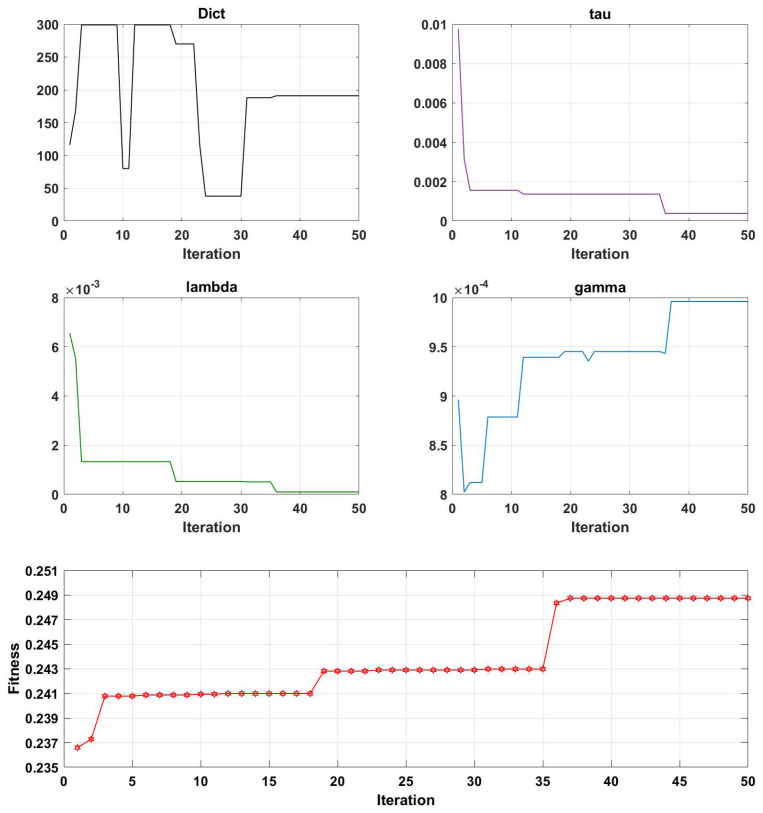
Change curves of the four parameters (*m*, τ, λ, and γ) and the fitness function on the MPED dataset. Here, ’Dict’ represents the *m* parameter.

**Figure 9 brainsci-13-00977-f009:**
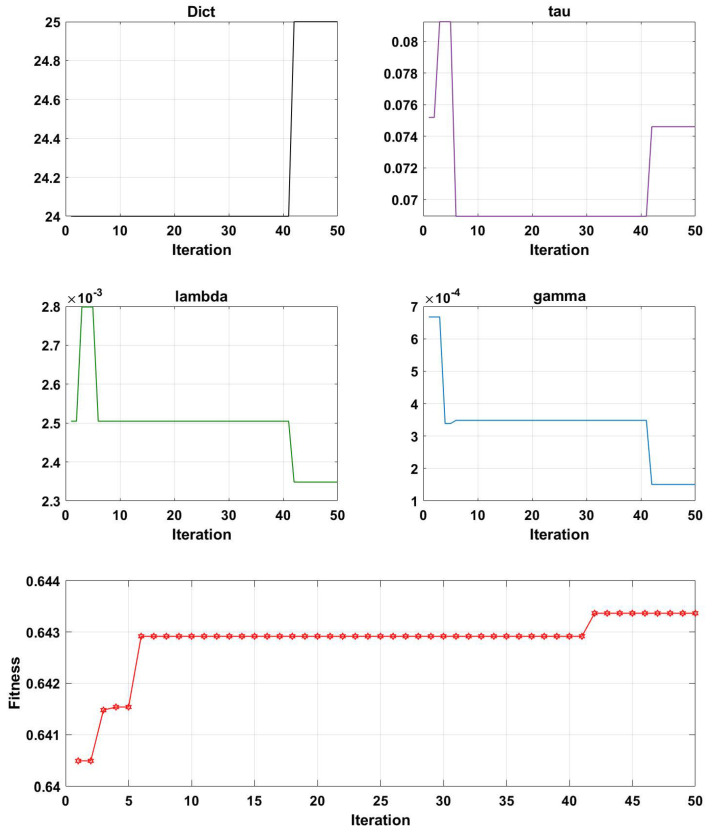
Change curves of the four parameters (*m*, τ, λ, and γ) and the fitness function on the GAMEEMO dataset (two—class). Here, ’Dict’ represents the *m* parameter.

**Figure 10 brainsci-13-00977-f010:**
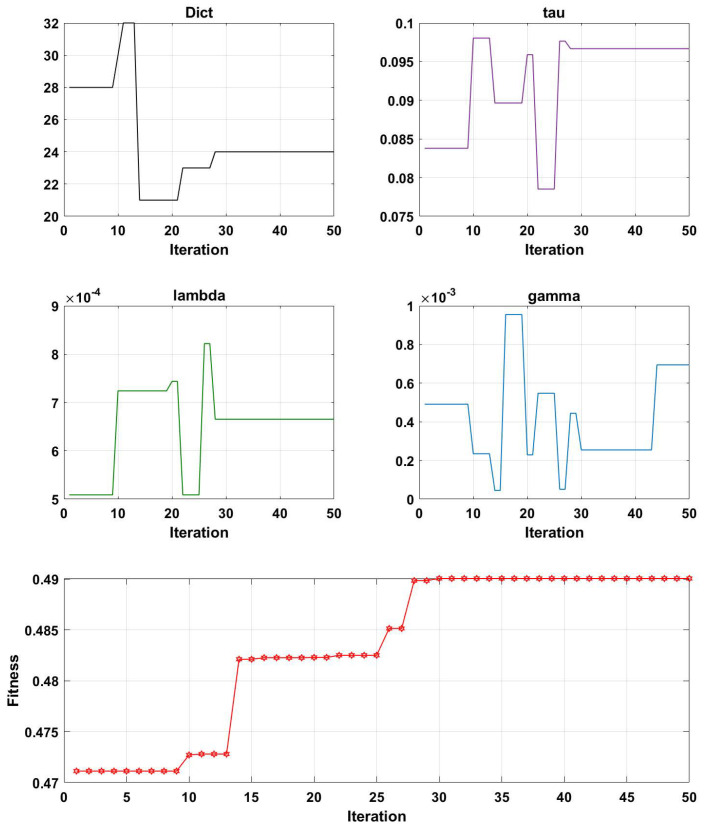
Change curves of the four parameters (*m*, τ, λ, and γ) and the fitness function on the GAMEEMO dataset (four—class). Here, ’Dict’ represents the *m* parameter.

**Figure 11 brainsci-13-00977-f011:**
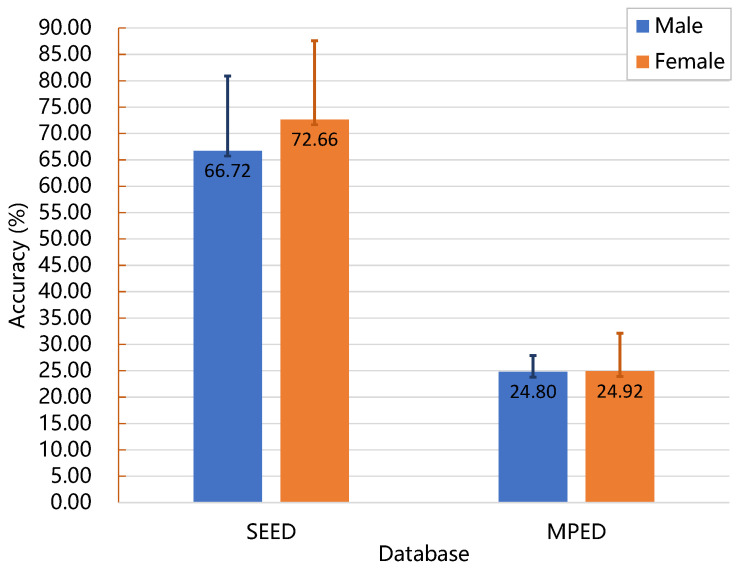
The mean and standard deviation of the accuracy for subjects of different sexes on the two databases.

**Table 1 brainsci-13-00977-t001:** The stimuli used in the GAMEEMO dataset.

Game Name	Stimuli Type	Positive–Negative	Arousal–Valence
G1	Boring	Negative	LANV
G2	Calm	Positive	LAPV
G3	Horror	Negative	HANV
G4	Funny	Positive	HAPV

HAPV: high arousal–positive valence; HANV: high arousal–negative valence; LANV: low arousal–negative valence; LAPV: low arousal–positive valence.

**Table 2 brainsci-13-00977-t002:** Initialization parameter settings of the GA.

Parameter	Value
Maximum generation	50
Size of population	20
Selection function	Stochastic Universal Sampling
Rate of individuals to be selected	0.9
Mutation probability	0.7

**Table 3 brainsci-13-00977-t003:** Matrix describing the length and how to decode each substring in the chromosome.

FieldD ${}^{1}$	*m*	τ	λ	γ
len ${}^{2}$	9	9	9	9
lb ${}^{3}$	1	0	0	0
ub ${}^{3}$	310/70 ${}^{*}$	0.1	0.01	0.001
code ${}^{4}$	gray	gray	gray	gray
scale ${}^{5}$	arithmetic	arithmetic	arithmetic	arithmetic
lbin ${}^{6}$	0	0	1	1
ubin ${}^{6}$	1	1	1	1

${}^{1}$ FieldD —Matrix describing the length and how to decode each substring in the chromosome. ${}^{2}$ len—row vector containing the length of each substring in the chromosome. sum(len) should equal the individual length. ${}^{3}$ lb, ub—Lower and upper bounds for each variable. ${}^{4}$ code—binary row vector indicating how each substring is to be decoded. ${}^{5}$ scale—binary row vector indicating where to use arithmetic and/or logarithmic scaling. ${}^{6}$ lbin, ubin—binary row vectors indicating whether or not to include each bound in the representation range. ${}^{*}$ 310 for SEED Dataset and MPED Dataset, 70 for GAMEEMO Dataset.

**Table 4 brainsci-13-00977-t004:** Mean accuracies (ACC) and standard deviation (STD) of all experiments.

Method	PDPL ACC/STD (%)	GA-PDPL ACC/STD (%)
SEED	51.02/13.57	69.89/14.39
MPED	21.39/5.41	24.87/5.83
GAMEEMO (two-class)	61.76/3.99	64.34/6.44
GAMEEMO (four-class)	39.92/8.28	49.01/8.46

**Table 5 brainsci-13-00977-t005:** The 95% confidence interval for the accuracy of all the experiments.

Method	PDPL	GA-PDPL
SEED	[43.51, 58.54]	[59.52, 76.78]
MPED	[19.05, 23.73]	[22.35, 27.39]
GAMEEMO (2-class)	[60.21, 63.31]	[61.84, 66.84]
GAMEEMO (4-class)	[36.71, 43.13]	[45.72, 52.29]

**Table 6 brainsci-13-00977-t006:** The training and testing time of all the experiments.

Method	PDPL		GA-PDPL	
**Time (s)**	**Training Time**	**Testing Time**	**Training Time**	**Testing Time**
SEED	5.6330	0.0208	76,902	0.001
MPED	12.6985	0.0625	335,386	0.005
GAMEEMO (2-class)	1.5734	0.0012	15,575	0.0005
GAMEEMO (4-class)	1.7660	0.0034	17,892	0.0004

**Table 7 brainsci-13-00977-t007:** Mean accuracies (M) and standard errors of the mean (SEM) of the subject-independent experiment on the SEED dataset (*N* = 15).

Method	M ± SEM (%)
KLIEP [40] *	45.17 ± 4.59
PDPL [27] *	51.02 ± 3.50
ULSIF [41] *	51.18 ± 3.50
STM [42] *	51.23 ± 3.83
SVM [43] *	56.73 ± 4.21
KPCA [44] *	61.28 ± 3.77
TCA [45] *	63.64 ± 3.84
SA [43] *	69.00 ± 2.81
GA-PDPL (ours )	**69.89 ± 3.72**

* indicates the experiment results obtained by our own implementation. Bold indicates best result.

**Table 8 brainsci-13-00977-t008:** Mean accuracies (M) and standard errors of the mean (SEM) of the subject-independent experiment on the MPED dataset (*N* = 23).

Method	M ± SEM (%)
KLIEP [40] *	18.92 ± 0.95
ULSIF [41] *	19.63 ± 0.79
TCA [45] *	19.50 ± 0.75
SVM [43] *	19.66 ± 0.83
GFK [49] *	20.27 ± 0.91
SA [43] *	20.74 ± 0.87
STM [42] *	20.89 ± 0.75
PDPL [27] *	21.39 ± 1.13
DANN [48]	22.36 ± 0.91
A-LSTM [16]	24.06 ± 0.96
GA-PDPL (ours )	**24.87 ± 1.22**

* indicates the experiment results obtained by our own implementation. Bold indicates best result.

**Table 9 brainsci-13-00977-t009:** Mean accuracies (M) and standard errors of the mean (SEM) of the subject-independent experiment on the GAMEEMO dataset (*N* = 28).

Method	Two-Class M ± SEM (%)	Four-Class M ± SEM (%)
KNN [46] *	58.16 ± 1.45	35.46 ± 2.06
Random Forest [47] *	59.29 ± 2.12	38.27 ± 2.95
PDPL [27] *	61.76 ± 0.75	39.92 ± 1.56
SVM [43] *	63.17 ± 1.27	46.62 ± 1.89
GA-PDPL(ours )	**64.34 ± 1.56**	**49.01 ± 1.60**

* indicates the experiment results obtained by our own implementation. Bold indicates best result.

## Data Availability

SEED Database at https://bcmi.sjtu.edu.cn/home/seed/seed.html, MPED Database https://github.com/Tengfei000/MPED, and GAMEEMO Database https://data.mendeley.com/datasets/b3pn4kwpmn/3, accessed on 18 June 2023.

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
