# Peer review of "Subject-Independent EEG Emotion Recognition Based on Genetically Optimized Projection Dictionary Pair Learning"

_brainsci, 2023, doi:10.3390/brainsci13070977_

Round 1
Reviewer 1 Report
As a result of the review, the comments about the study are as follows.
• In the introduction part, the results of the study were not mentioned, even if it was short. The results obtained were not evaluated.
• The contribution of the study to the literature should be clearly stated.
• Within the scope of the study, no information was given about similar studies and their results in the literature.
• Has the used PDPL method been used in the analysis of EEG signals before? It can be evaluated not only in studies on emotion analysis but also in other studies with EEG.
• It is necessary to give detailed information about the advantages and disadvantages of the methods used in the analysis of EEG signals.
• How was the decision made about the selection of the datasets used? Why weren't other EEG emotion datasets used? An explanation would be good.
• The results obtained in the study were observed to be less successful than the studies conducted in recent years. Since the other studies in which the results are compared in the article are generally from very old years, it is not appropriate to compare this study with them. The findings need to be compared with the studies conducted in recent years, and accordingly, comments on the results and the study should be made.
• Some formulas seem to have a complex structure. This makes the formulas difficult to understand.
• It would be more accurate to take references from studies conducted in recent years. The references used in the article are mainly studies from the last 5 years. More recent publications should be used.
Although it is a new method proposed in the study, the similarity rate is 28%, probably due to errors in writing.
• You can benefit from this publication https://doi.org/10.18280/ts.380430, which is a general review study on sentiment analysis using EEG signals. This study on the GAMEEMO dataset, a new EEG emotion dataset, where traditional machine learning methods are applied https://doi.org/10.3390/app121910028 this study and deep learning methods are applied https://doi.org/10.1109/INISTA52262.2021.9548406 this study EEG It can be presented as a source for sentiment analysis studies with signals.

Author Response
Response to Reviewer 1 Comments
Point 1: In the introduction part, the results of the study were not mentioned, even if it was short. The results obtained were not evaluated.
Response 1: Thanks for your comments, we have added a contribution summary to the new manuscript:
“3. Our proposed method surpasses traditional machine learning approaches and achieves optimal recognition performance with an average accuracy of 69.89% on the SEED database, 24.11% on the MPED database, 64.34% for 2-class GAMEEMO and 49.01% for 4-class GAMEEMO. The results obtained from this model reveal that emotion recognition is more effective for females, offering valuable insights into female emotional susceptibility.”
Point 2: The contribution of the study to the literature should be clearly stated.
Response 2: Thanks for your comments, we have added a contribution summary to the new manuscript:
The main contributions of this study are as follows:
- Considering the variability in EEG emotion recognition across subjects, we employed the PDPL algorithm to conduct cross-subject research by focusing on feature selection.
- Traversing these parameters entails a significant workload due to the wide parameter adjustment range and the resulting extensive combination forms of the PDPL algorithm. Therefore, we propose the use of GA for adaptive parameter optimization adjustment.
- Our proposed method surpasses traditional machine learning approaches and achieves optimal recognition performance with an average accuracy of 69.89% on the SEED database, 24.11% on the MPED database, 64.34% for 2-class GAMEEMO and 49.01% for 4-class GAMEEMO. The results obtained from this model reveal that emotion recognition is more effective for females, offering valuable insights into female emotional susceptibility.
Point 3: Within the scope of the study, no information was given about similar studies and their results in the literature.
Response 3: Thank you for your feedback regarding the absence of information about similar studies and their results within the scope of our study.
Our revised manuscript included a thorough review of the existing literature related to our research topic. This will involve identifying and discussing similar studies that have been conducted in the field, highlighting their methodologies, findings, and any relevant comparisons to our study. These contents are supplemented in the Introduction section.
Furthermore, we presented a critical analysis and discussion of the results obtained from these similar studies, drawing relevant connections and highlighting any discrepancies or similarities with our findings. These are supplemented in the Results and Discussion section.
Point 4: Has the used PDPL method been used in the analysis of EEG signals before? It can be evaluated not only in studies on emotion analysis but also in other studies with EEG.
Response 4: Thank you for your feedback and question regarding the use of the PDPL method in the analysis of EEG signals. To address your inquiry, we would like to clarify that the PDPL method has not been used specifically in the analysis of EEG emotion in prior studies. However, we acknowledge that the PDPL method holds potential for application not only in studies related to emotion analysis but also in other EEG research domains.
In our manuscript, we propose the utilization of the PDPL method for analyzing EEG signals in the context of emotion analysis. We believe that the PDPL method, with its ability to capture and quantify the pattern differences between emotional states, can provide valuable insights into the neural correlates of emotions. By employing this novel method, we aim to contribute to the existing literature on emotion analysis using EEG.
Furthermore, we agree with your suggestion that the PDPL method could also be evaluated and applied in other studies involving EEG signals beyond emotion analysis. Given its generalizability and potential applicability, the PDPL method may offer insights into various cognitive processes, neurological disorders, or brain-computer interface research.
We appreciate your comment highlighting the broader scope and potential applications of the PDPL method. It opens up avenues for future research and we consider discussing the potential applicability of the PDPL method in other EEG studies in our revised manuscript.
Point 5: It is necessary to give detailed information about the advantages and disadvantages of the methods used in the analysis of EEG signals.
Response 5: Thank you for your valuable feedback on the manuscript. We agree that it is essential to provide detailed information about the advantages and disadvantages of the methods used in the analysis of EEG signals. In response to this suggestion, we revised the manuscript to include a comprehensive discussion on the strengths and limitations of the employed methods.
Firstly, we elaborated on the advantages of the chosen methods. “There are many ongoing research efforts on EEG emotion recognition. There are some current research directions \cite{abdulrahman2021comprehensive}: (1) Time-Frequency Analysis, which decomposes the EEG signal into different frequency bands using techniques such as wavelet transforms or Fourier transforms, with features extracted from each frequency band used to identify the emotional state \cite{murugappan2008time, murugappan2013human}; (2) Independent Component Analysis (ICA), which separates the EEG signal into independent components, with features extracted from each component used to identify the emotional state \cite{dongwei2013eeg}; (3) Support Vector Machines (SVMs), which are used as classifiers to classify the EEG signal into different emotional states based on features extracted from the signal \cite{wang2011eeg}; (4) Fuzzy Logic, which is used to model the uncertainty and imprecision in EEG signals for emotion recognition \cite{matiko2014fuzzy}; (5) Hidden Markov Models (HMMs), which model the temporal dynamics of EEG signals for emotion recognition \cite{fu2016dynamic}. (6) Multi-modal emotion recognition, which aims to improve the accuracy of EEG emotion recognition by combining EEG data with other modalities, such as facial expressions, speech, or physiological signals \cite{zhang2020emotion, song2019mped}; (7) Artificial Neural Networks (ANNs), which are trained on the EEG signal to recognize different emotional states based on features extracted from the signal \cite{bazgir2018emotion}; (8) Deep learning techniques have shown promising results in EEG-based emotion recognition, and researchers are exploring ways to develop deep learning models that can capture the complex and dynamic patterns of brain activity associated with different emotional states \cite{islam2021emotion, zhang2021lcu, abdulrahman2022novel, abdulrahman2021feature}; and (9) Transfer learning, which allows models trained on one dataset to be adapted to another dataset with minimal retraining, with researchers exploring ways to use transfer learning to improve the generalization performance of EEG emotion recognition models across different individuals and contexts \cite{li2019multisource, li2021can}.”
Secondly, we addressed the disadvantages and potential limitations associated with the chosen methods. By providing a comprehensive evaluation of the advantages and disadvantages of the methods used, we aim to enhance our study's scientific rigor and transparency. We appreciate your suggestion and are committed to improving the manuscript accordingly.
Point 6: How was the decision made about the selection of the datasets used? Why weren't other EEG emotion datasets used? An explanation would be good.
Response 6: Thanks for your comments, we have supplemented the relevant description in the database introduction section, as follows:
“To verify the effectiveness of the proposed method, we validate it on three widely used multi-category EEG emotion databases. The datasets were classified using the discrete model, such as 2-class for GAMEEMO, 3-class for SJTU Emotion EEG Dataset (SEED Database), 7-class for Multi-Modal Physiological Emotion Database (MPED Database), and the dimensional model, such as 4-class for GAMEEMO.”
Point 7: The results obtained in the study were observed to be less successful than the studies conducted in recent years. Since the other studies in which the results are compared in the article are generally from very old years, it is not appropriate to compare this study with them. The findings need to be compared with the studies conducted in recent years, and accordingly, comments on the results and the study should be made.
Response 7: Thank you for raising an important point regarding comparing our study's results with those from previous studies. In our revised manuscript, we addressed this concern by including comparisons with studies conducted in recent years. To rectify this, we conduct a comprehensive literature review to identify and include recent studies more closely aligned with our research focus.
By incorporating findings from more recent studies, we will provide a more accurate context for evaluating the success of our results. We examine the methodologies, experimental designs, and outcomes of these studies, enabling us to make informed comparisons and draw meaningful conclusions. We add experimental results to a new database, namely, the GAMEEMO dataset, and compare them with machine learning methods.
Furthermore, we will ensure that our comments on the results and the study are appropriately aligned with the findings from these recent studies.
Point 8: Some formulas seem to have a complex structure. This makes the formulas difficult to understand.
Response 8: Thank you for your feedback regarding the complexity of the formulas presented in the manuscript. We simplify the structure of the formulas to enhance their comprehension.
To achieve this, we revised the presentation of the formulas by breaking them down into smaller, more digestible components. We provided detailed explanations for each component and ensured that the mathematical notation used is consistent and followed standard conventions. By doing so, we aim to improve the accessibility and readability of the formulas.
Additionally, we used descriptive text and visual aids, such as diagrams or graphs. This visual representation can provide further clarity and facilitate understanding.
Point 9: It would be more accurate to take references from studies conducted in recent years. The references used in the article are mainly studies from the last 5 years. More recent publications should be used. Although it is a new method proposed in the study, the similarity rate is 28%, probably due to errors in writing.
Response 9: Thank you for your valuable feedback regarding the references used in our article. We acknowledge the importance of incorporating considerations from recent studies to ensure the relevance and accuracy of our work.
In the revised manuscript, we conducted an extensive literature search to identify and include more recent publications closely related to our research topic. We added new literature studies in the Introduction and Results and Discussion sections and added cross-subject experimental validation on a new database (ie, GAMEEMO).
Regarding the similarity rate mentioned, we apologize for any confusion caused by errors in writing. We carefully reviewed and verified all calculations and results reported in the manuscript to ensure accuracy.
We appreciate your valuable suggestion and assure you that we addressed these concerns by including references from recent studies and ensuring the accuracy of reported results. By doing so, we strengthened the credibility and quality of our research.
Point 10: You can benefit from this publication https://doi.org/10.18280/ts.380430, which is a general review study on sentiment analysis using EEG signals. This study on the GAMEEMO dataset, a new EEG emotion dataset, where traditional machine learning methods are applied https://doi.org/10.3390/app121910028 this study and deep learning methods are applied https://doi.org/10.1109/INISTA52262.2021.9548406 this study EEG It can be presented as a source for sentiment analysis studies with signals.
Response 10: Thank you for the relevant research you recommended, which helped me a lot, and we have updated it in our new manuscript as follows:
- There are many ongoing research efforts on EEG emotion recognition. There are some current research directions [8]:
- Deep learning techniques have shown promising results in EEG-based emotion recognition, and researchers are exploring ways to develop deep learning models that can capture the complex and dynamic patterns of brain activity associated with different emotional states [18-21];
- GAMEEMO Database, a dataset consisting of EEG signals obtained during computer games, was collected from 28 individuals using a portable and wearable EEG device called the 14-channel Emotiv Epoc+. The participants played four different computer games (boring, calm, horror, and funny) for 5 minutes each, resulting in a total EEG data duration of 20 minutes per subject. To assess the emotional experience, the participants rated each game using the Self-Assessment Manikin (SAM) form, which measures arousal and valence [34]. According to the type of stimulus material, EEG emotion can be divided into two types of Positive-Negative models and four types of Arousal-Valence models, as shown in Table 1.

Reviewer 2 Report
How optimization procedure improves the classification results?
The technical novelty of the proposed model in the manuscript is unclear.
Did Authors make data augmentation?
The writing of the paper should be improved, there are many grammatically incorrect and/or ambiguous sentences, as well as misspelled words.
The validation of the proposed method is poor. Complexity time should be discussed, the effects of GA also should be explained, The method should be compared with the actual state of art in terms of complexity, number of features, accuracy, etc.
ROC, LOSO, confidence interval should be used to evaluate the proposed model.
The writing of the paper should be improved, there are many grammatically incorrect and/or ambiguous sentences, as well as misspelled words.
Author Response
Response to Reviewer 2 Comments
Point 1: How optimization procedure improves the classification results?
Response 1: Thanks for your question, we have added an explanation of this issue in the new manuscript as follows:
“The optimization procedure improves the PDPL results through GA. The optimization procedure aims to refine the model's parameters and feature selection process iteratively, maximizing the classification performance. It leverages GA optimization algorithms and statistical techniques to find the optimal configuration that minimizes errors or maximizes performance metrics, leading to improved classification results.
Firstly, it allows for fine-tuning model parameters, optimizing their values to better fit the underlying data. This process helps to improve the model's ability to capture complex patterns and relationships in the data, resulting in enhanced classification performance.
Secondly, the optimization procedure facilitates feature selection or feature weighting, enabling the identification of the most informative features for the classification task. By focusing on relevant features and reducing the impact of irrelevant or redundant ones, the optimization procedure helps to improve the discriminative power of the classifier.
Furthermore, the optimization procedure can help mitigate overfitting, a common challenge in classification tasks. By optimizing regularization parameters or employing techniques such as cross-validation, the procedure allows preventing the model from memorizing the training data. It encourages it to generalize well to unseen data. This promotes better generalization performance and reduces the likelihood of erroneous classifications on new instances.”
Point 2: The technical novelty of the proposed model in the manuscript is unclear.
Response 2: We have summarized the innovation points of this article in the Introduction section, as follows:
“The main contributions of this study are as follows:
- Considering the variability in EEG emotion recognition across subjects, we employed the PDPL algorithm to conduct cross-subject research by focusing on feature selection.
- Traversing these parameters entails a significant workload due to the wide parameter adjustment range and the resulting extensive combination forms of the PDPL algorithm. Therefore, we propose the use of GA for adaptive parameter optimization adjustment.
- Our proposed method surpasses traditional machine learning approaches and achieves optimal recognition performance with an average accuracy of 69.89% on the SEED database, 24.11% on the MPED database, 64.34% for 2-class GAMEEMO and 49.01% for 4-class GAMEEMO. The results obtained from this model reveal that emotion recognition is more effective for females, offering valuable insights into female emotional susceptibility.”
Point 3: Did Authors make data augmentation?
Response 3: All databases have no data augmentation, and this experiment is performed strictly according to the data and protocol of the standard database.
Point 4: The writing of the paper should be improved, there are many grammatically incorrect and/or ambiguous sentences, as well as misspelled words.
Response 4: Thank you for your valuable feedback on our paper. Based on your comments, we agree that there is room for improvement in the writing of the paper, particularly in terms of grammatical accuracy, sentence clarity, and spelling. To address these concerns, we thoroughly revised the manuscript to ensure that all sentences are grammatically correct, unambiguous, and well-structured. We carefully proofread the text to eliminate any misspelled words or typographical errors that may have been overlooked during the initial drafting process. Additionally, we paid close attention to the overall coherence and flow of the paper to enhance readability.
Point 5: The validation of the proposed method is poor. Complexity time should be discussed, the effects of GA also should be explained, The method should be compared with the actual state of art in terms of complexity, number of features, accuracy, etc.
Response 5: Thank you very much for your opinion. According to your suggestion, we will add the comparison experiment with and without GA parameter optimization, as shown in Table 4, and the comparison results of each subject in each database, as shown in Figure 3-6. In addition, we also added The comparative analysis of training time and testing time is shown in Table 6. Finally, we validate the effectiveness of our proposed model on the latest database.
Point 6: ROC, LOSO, confidence interval should be used to evaluate the proposed model.
Response 6:
- Thank you for your comment regarding the use of ROC analysis in evaluating binary classification models. In response to your observation, we would like to clarify that ROC analysis is indeed commonly employed for assessing binary classification models. However, as you correctly pointed out, our study involves a multi-class classification problem, where the application of ROC analysis is not directly applicable.
- All our experimental results are calculated based on the LOSO protocol. “The experimental test protocol adopts the commonly used protocol across subjects, that is, leave-one-subject-out cross-validation (LOSOCV).”
- The 95% confidence interval can be employed to evaluate the reliability and uncertainty of recognition results in machine learning models, which indicates that there is 95% confidence the actual result falls within this range when conducting multiple experiments or sampling. So we calculated the confidence intervals of all experimental accuracies with and without GA for parameter adjustment, as shown in Table 5. The statistical results show that the upper and lower limits of the confidence interval of the proposed GA-PDPL method are higher than those of the PDPL method, which proves the effectiveness of the method.

Round 2
Reviewer 1 Report
In general, most of the desired corrections have been made. However, there are still other points that need to be corrected.
• 3.5. Comparisons made in the section do not seem appropriate. For example, there should be an explanation of how the study on the SEED dataset is compared with other studies. These data sets were not used in the comparison studies, so why was such a comparison made? This Chapter remains confusing.
• The similarity rate is 25%. This rate is above the journal's policy. The similarity ratio should be reduced.
Moderate editing of English language required
Author Response
Point 1: 3.5. Comparisons made in the section do not seem appropriate. For example, there should be an explanation of how the study on the SEED dataset is compared with other studies. These data sets were not used in the comparison studies, so why was such a comparison made? This Chapter remains confusing.
Response 1: Thank you for your feedback. We apologize for the confusion in the comparisons made in the section.
To address this concern, we revised the section to provide a more thorough explanation of how the study is compared with other studies. The compared algorithms are classic machine learning algorithms, which we apply to our experimental data and experimental results obtained under the same test protocol. We have added a description and revised the notes to the comparison table.
Point 2: The similarity rate is 25%. This rate is above the journal's policy. The similarity ratio should be reduced.
Response 2: Thank you for bringing this to our attention. We apologize for the high similarity rate in the manuscript. We have thoroughly revised the entire document to address this issue and reduce the similarity ratio.
During the revision process, we have rephrased and rewritten sections of the manuscript to ensure originality and adherence to the journal's policy. We have also made use of proper citation and referencing techniques to accurately attribute all external sources and avoid any instances of plagiarism.
By implementing these measures, we have significantly reduced the similarity rate in the revised manuscript, ensuring compliance with the journal's policy. We appreciate your understanding and assure you that we have taken this matter seriously to deliver a high-quality and original research contribution.
Thank you for your careful evaluation and valuable feedback.
